# Current Treatments of Metastatic Colorectal Cancer with Immune Checkpoint Inhibitors—2020 Update

**DOI:** 10.3390/jcm9113520

**Published:** 2020-10-31

**Authors:** Gerhard Jung, Daniel Benítez-Ribas, Ariadna Sánchez, Francesc Balaguer

**Affiliations:** 1Gastroenterology Department, Hospital Clínic de Barcelona, Centro de Investigación Biomédica en Red de Enfermedades Hepáticas y Digestivas (CIBERehd), Institut d’Investigacions Biomèdiques August Pi i Sunyer, University of Barcelona, 08036 Barcelona, Spain; JUNG@clinic.cat (G.J.); ASANCHEZG@clinic.cat (A.S.); 2Colorectal Cancer Research Laboratory, Institute for Research in Biomedicine, Parc Científic de Barcelona, 08028 Barcelona, Spain; 3Immunology Department, Centre for Biomedical Diagnosis, Hospital Clínic de Barcelona, Institut d’Investigacions Biomèdiques August Pi i Sunyer, 08036 Barcelona, Spain; DBENITEZR@clinic.cat

**Keywords:** colorectal cancer, metastatic colorectal cancer, advanced colorectal cancer, treatment refractory colorectal cancer, immunotherapy, checkpoint inhibitors, pembrolizumab, nivolumab, atezolizumab, ipilimumab

## Abstract

During the last 20 years, chemotherapy has improved survival rates of colorectal cancer (CRC). However, the majority of metastatic cases do not respond to or progress after first line conventional chemotherapy and contribute to the fatalities of patients with CRC. Insights into the immune contexture of the tumor microenvironment (TME) have enabled the development of new systemic treatments that boost the host immune system against the tumor—the immune checkpoint inhibitors (ICI). These promising drugs have already shown astonishing efficacies in other cancer types and have raised new hope for the treatment of metastatic CRC (mCRC). In this review, we will summarize the results of the clinical trials that led to their accelerated approval by the U.S. Food and Drug Administration (FDA) in 2017, as well as all relevant recent studies conducted since then—some of which are not published yet. We will focus on therapeutic efficacy, but also discuss the available data for drug safety and security, changes in quality of life indicators and predictive biomarkers for treatment response. The burgeoning evidence for a potential use of ICIs in other settings than mCRC will also be mentioned. For each trial, we have made a preliminary assessment of the quality of clinical trial design and of the “European Society of Medical Oncology (ESMO) magnitude of clinical benefit” (ESMO-MCBS) in order to provide the first evidence-based recommendation to the reader.

## 1. Introduction

Cancer immunotherapy aims to enhance the natural capability of the immune system to fight cancer cells and has already become one of the pillars of cancer treatment in advanced stages [1]. Immune checkpoint inhibitors (ICI) have demonstrated ground-breaking results in tumors with a high burden of genetic mutations such as melanoma or lung cancer [2]. Due to their high mutagenic level, these tumors generate many neoantigens and provoke a strong immunogenic reaction driven by T-cells. Programmed death cell protein 1 (PD-1) is expressed on the surface of these T-cells and interacts with programmed death-ligand 1 or 2 (PD-L1, PD-L2), leading to a suppression of the immune response by transmitting an inhibitory signal to the cytotoxic T-cells and reducing apoptosis in regulatory T-cells. Cancers use this mechanism to evade the immune response by over-expressing programmed cell death ligand-1/2 (PD-L1/2) on their cell surface [3]. In a similar way, cytotoxic T-lymphocyte-associated protein 4 (CTLA-4) receptors are up-regulated on the surface of activated T-lymphocytes and compete with CD28 receptors for the ligands CD80/86 expressed on antigen presenting cells (APCs). While CD28 is a co-activating factor for T-cells, CTLA-4 sends an inhibitory signal to T-cells and outcompetes CD28 as it binds with higher affinity and avidity to its ligands CD80/86. Immune checkpoint inhibitors (ICI) are monoclonal antibodies that block these pathways by binding to the PD-1 receptor (i.e., nivolumab, pembrolizumab), to PD-L1 (i.e., atezolizumab), or to CTLA-4 (i.e., ipilimumab) and thus enhance the immune response against cancer cells.

Recent evidence indicates that in metastatic colorectal cancer (mCRC) patients, ICIs’ response is limited to those with high mutational burden showing mismatch repair deficiency (MMRd). Professional organizations recommend testing all newly diagnosed CRCs for MMRd either by immunohistochemistry (IHC) to detect loss of expression of the mismatch repair (MMR) proteins (MLH1, MSH2, MSH6 or PMS2) or by polymerase chain reaction (PCR) or next generation sequencing (NGS) of microsatellite instability (MSI) markers [4,5,6,7,8]. MMRd CRC tumors have a high mutational load (and specially frame-shift mutations) that creates many neoantigens which are presented on major histocompatibility complex (MHC) molecules and are recognized as foreign by T-cells. As a consequence, MMRd tumors have much higher PD-L1 expression in tumoral cells and tumor associated macrophages (TAMs) and a higher presence of tumor-infiltrating lymphocytes (TIL) than MMR proficient (MMRp) tumors. This subtype of CRC accounts for approximately 5% of all mCRC and has shown an impressive benefit of treatment with ICIs, which led to their accelerated approval by the U.S. Food and Drug Administration (FDA) in 2017 [9,10].

In this review article, we will give an overview of the results of phase II and III clinical trials including unpublished data, with a clear focus on treatment efficacy and safety (Table 1). Large clinical controlled trials are still scarce, and many clinicians are still unfamiliar with these novel drugs. In order to provide the first graded recommendation, we aimed to interpret the current results of each of the trials by carefully assessing the quality of design and the European Society of Medical Oncology (ESMO) magnitude of clinical benefit (as described in Cherny NI et al., 2017) [11]. Our recommendations succumb to upcoming evidence of ongoing studies but shall help clinicians make therapeutic decisions and to encourage or discourage enrolling in new trials.

## 2. Treatment Efficacy

### 2.1. Pembrolizumab (Anti PD-1) for Treatment Refractory mCRC

The first phase II study of pembrolizumab for the treatment of mCRC patients was published by Le and colleagues in 2015 in the New England Journal of Medicine (NEJM) [12]. This study comprised three cohorts consisting of 11 MMRd CRCs, 21 MMRp CRCs and nine MMRd non-colorectal gastrointestinal (GI) cancers. All patients with CRCs had had at least two or more previous chemotherapy regimens. Since the MMRd mCRC cohort included nine cases of Lynch syndrome, their age was somewhat younger compared to the other groups (46 y vs. 61 y and 57 y, *p* < 0.001). The results for the primary endpoint objective response rate (ORR), evaluated by ‘Response evaluation criteria in solid tumors’ (RECIST) v1.1, were 40% (95% confidence interval (CI) 12–74) for the MMRd mCRC cohort, while the ORR for the MMRp cohort was 0%. Although no complete response (CR) was observed, the disease control rate (DCR) was 90%, consisting of 40% partial responses (PR) and 50% stable disease (SD) at 12 weeks. After a median follow up of 36 weeks in the MMRd mCRC cohort and 20 weeks in the MMRp cohort, the median progression free survival (PFS) as a co-primary endpoint for the MMRd cohort was not reached while it was only 2.2 months for the MMRp group, with a hazard ratio (HR) of 0.10 (*p* < 0.001). Progression free survival rates at 20 weeks for both cohorts were 78% and 11%, respectively. Median overall survival (OS) was also not reached in the MMRd group, while it was 5.0 months for MMRp mCRCs.

In 2017, Le and colleagues [13] published an updated trial with pembrolizumab that comprised 86 patients with twelve different treatment refractory progressive MMRd cancers, 40 of which were MMRd mCRCs. The disease control rate was slightly lower compared to their previous study but showed a remarkable complete response rate of 12%. The average time to any response was 21 weeks and to complete response 42 weeks. Again, neither the median progression free survival nor the overall survival was reached, with an estimate at one year of 64% and 76%, respectively. Both progression free survival (PFS) and overall survival (OS) were similar in the mCRC and non-CRCs cohorts. They also found no significant differences between patients with a diagnosis of Lynch syndrome and no-Lynch syndrome.

In 2017, Díaz and colleagues presented the results of the Keynote 164 trial on the European Society of Medical Oncology (ESMO) conference (not published) [14]. In this study, 61 patients with MMRd mCRCs were treated with 200 mg of pembrolizumab every three weeks and response was assessed every nine weeks. As a primary endpoint, the ORR was 27.9% (95% CI 17.1–40.8) and PFS and OS rates at six months were 43% and 87%, respectively. Of note, the Keynote 158 trial was analyzed in the same manner (77 microsatellite instability high (MSI-H) non-CRCs) and came to similar results: ORR 37.7%, PFS at six months 45% and OS at six months 73%.

Finally, in 2020, the updated results of the Keynote 164 trial were published [15], in which they separately analyzed a cohort of 61 patients who had received more previous lines of therapy (10% only one previous line and 90% ≥2) and a second cohort of 63 patients who had received less previous lines of therapy (38% only one previous line and 62% ≥2). All tumors were MMRd and/or MSI-H, unresectable, locally advanced or metastatic CRCs. The primary endpoint ORR was equal in both cohorts with 33% (95% CI 21–46) though the rate of complete responses was slightly higher in the less pretreated cohort, 7.9% vs. 3.3%, respectively. The median PFS was also higher in cohort two with 4.1 months vs. 2.3 months, with an estimated 12 month PFS rate of 41% vs. 34%, respectively. The median overall survival was not reached in the less pretreated cohort, while it was 31.4 months in the group with more previous lines of treatment, with an estimated 12 month OS rate of 76% and 72%, respectively. In summary, this update confirmed the durable clinical benefit in pretreated patients, though PD-1 blockers may be more effective in earlier disease stages and no new safety signals were identified.

Quality and recommendation: numerically, the quality of clinical trial design results was low for all studies, since there were no adequate control arms. However, the prespecified objectives were achieved in all of them and there is a strong clinical benefit in terms of increase in overall survival (ESMO-MCBS 2/3). Strong recommendation in favor of the use of pembrolizumab in refractory MMRd, locally advanced unresectable or metastatic CRC.

### 2.2. Pembrolizumab as First Line Treatment for MMRd mCRC

Recently, in May 2020, Andre and colleagues presented the interim analysis of the open-label, randomized Keynote-177 trial [16], which evaluated the efficacy and safety of pembrolizumab as a first line treatment versus standard of care chemotherapy (SOC) in MMRd mCRC. Strikingly, the pembrolizumab cohort showed a clinically meaningful and statistically significant improvement of the primary endpoint median PFS (16.5 vs. 8.2 months, hazard ratio (HR) 0.6, *p* = 0.0002) with almost half of the patients (48.3%) in the pembrolizumab arm free of progression at data cut-off (24 months), while presenting less grade 3–5 drug related adverse events than in the standard of care (SOC) chemotherapy arm (22% vs. 66%). The evaluation of the overall survival as a co-primary endpoint is still ongoing.

Quality and recommendation: this trial was conducted with an adequate control arm that consisted of mFOLFOX6 or FOLFIRI ± anti-EGFR (cetuximab) or anti-VEGF (bevacizumab), chosen by the investigator before randomization. There were no changes in primary endpoints or sample size and the prespecified objectives were achieved. Thus, the clinical trial design was of high quality and in view of the convincing clinical benefit, pembrolizumab is highly recommended as a first line treatment option for MMRd mCRC.

### 2.3. Nivolumab (Anti-PD-1) in Monotherapy and Combination of Nivolumab and Ipilimumab (Anti-CTLA-4) for Treatment Refractory MMRd mCRC

The Checkmate 142 study [17] demonstrated for the first time a durable response and disease control with nivolumab in heavily pre-treated MMRd mCRC patients. In the first step, nivolumab was administered as monotherapy to 74 patients, 85% of which had failed two or three previous lines of treatment. The primary end point was the ORR evaluated by RECIST v1.1 in a local and centralized manner. Centralized data showed an ORR of 32.4% (95% CI 22.0–44.3), with 2.7% complete and 29.7% partial response and were only slightly different to local data evaluation (31.1%). Secondary end points were duration of response whose median was not yet reached, progression free survival at one year 50.4% (95% CI (38–61)) and overall survival at one year 73.4% (95% CI (62–82)). Remarkably, 63.5% achieved a disease control defined as complete response, partial response or stable disease of 12 weeks or longer (95% CI (57.5–74.4)).

As a second step, Overman and colleagues [18] analyzed the efficacy of combining nivolumab with ipilimumab. Nivolumab plus ipilimumab was administered to 119 patients with MMRd mCRCs, 76% of whom had received two or three lines of previous treatment. The ORR as a primary endpoint reached 49% (95 CI% (39.5–58.1)) with 4% complete responses and 45% partial responses. The median PFS and OS were not reached and estimates at one year were 71% (95% CI (61.4–78.7)) and 85% (95% CI (77.0–90.2)), respectively. Disease control for 12 weeks or longer and durable response for 12 months or longer were 79% and 83%, respectively. In conclusion, all endpoints showed better results compared to nivolumab alone, while the median time to response was similar in both cohorts (2.8 months).

Quality and recommendation: both trials are classified as low quality in clinical trial design, basically due to the absence of an adequate control arm. However, they both achieved their pre-specified objective and a strong clinical benefit in terms of increase in overall survival. Strong recommendation in favor of nivolumab for MMRd mCRC after prior lines of therapy. Moderate recommendation for adding ipilimumab, which showed slightly better response rates while presenting a similar safety profile.

### 2.4. Nivolumab–Ipilimumab Combination Therapy as Neoadjuvant Treatment for Early Stage CRC

In a recent preliminary phase Ib trial, Chalabi and colleagues [19] administered nivolumab combined with ipilimumab to 20 patients with MMRd and to 15 patients with MMRp early stage (I, II or III) CRC as a neoadjuvant treatment before surgery (nivolumab 3 mg/kg on day one and day 15, and ipilimumab 1 mg/kg on day one). Moreover, eight MMRp patients were randomized to receive also celecoxib (200 mg every day from day one until surgery). Furthermore, 100% of MMRd patients showed a pathological response (95% CI (86–100)), with 60% complete responses and 95% major responses (<10% residual viable tumor). However, more strikingly, 27% (95% CI (8–55)) of MMRp tumors also showed a pathological response, with 13.3% complete responses, 20% major pathological responses (<10% viable tumor) and 6.7% partial responses. During a median follow-up of nine months (interquartile range (IQR) 5.3–15.7), only one patient of the MMRp cohort, who was staged T3N1 and did not show response to ICI, developed a liver metastasis that required metastasectomy. Another MMRp patient died due to a cardiovascular event that was not drug related. The rest of the patients were all alive at data cutoff and all underwent surgery during the pre-specified period (maximum six weeks after inclusion). MMRp patients who were randomized to also receive celecoxib did not improve responses more than those that were not. Of note in this study, not only responders but also MMRp non-responders showed a significant increase in CD8+T-cells, T-cell receptor (TCR) clonality and other immune scores though they did not lead to tumor regression, and CD8+/PD-1+T-cell infiltration was predictive of response in these tumors.

Quality and recommendation: this study provides the first evidence of efficacy for immune checkpoint inhibitors in a neoadjuvant setting in early stage MMRd CRC and a subgroup of MMRp CRC, and CD8+/PD1+T-cell infiltration was identified as a potential biomarker for response. Due to the small sample size and absence of an adequate control arm, these promising findings should be confirmed in future well designed studies before a clear recommendation can be made.

### 2.5. Atezolizumab (Anti-PD-L1) and Cobimetinib (Anti-MAPK) as Third Line for Chemo-Resistant mCRC

Regorafenib is a multi-kinase inhibitor and is actually the standard of care (SOC) as a third line therapy for chemo-resistant mCRC, but achieves a PFS of less than two months (HR 0.49, *p* < 0.0001 (vs. placebo)) and an OS of hardly more than six months (HR 0.77, *p* = 0.0052 (vs. placebo)) [20]. Based on the findings from preclinical studies that found that mitogen-activated protein kinase (MAPK) signaling is involved in mechanisms of immune evasion, particularly down-regulation of MHC-I molecules and up-regulation of immunosuppressive cytokines, it was hypothesized that ICIs and MAPK inhibitors could act synergistically [21]. After favorable outcomes in terms of responses and safety from early phase studies, Eng and colleagues [22] enrolled 363 patients in a large phase III trial (named IMblaze370) in which they compared atezolizumab—an anti-PD-L1 monoclonal antibody—in combination with cobimetinib, a MEK1/2 inhibitor (cohort one, 183 patients), or atezolizumab monotherapy (cohort two, 80 patients) with standard of care regorafenib (cohort three, 80 patients) for unresectable, locally advanced CRC or mCRC with at least two prior treatments. In this trial, MMRd tumors were capped at 5% during enrollment. Unfortunately, no complete responses in any of the cohorts and very low rates of partial responses with no differences between the cohorts (2–3%) were observed. There were also no differences in PFS (1.91, 1.94 and 2.0 months, respectively) nor OS (8.9, 7.1 and 8.5 months, respectively) between the three treatment arms.

Quality and recommendation: the IMblaze370 trial was well designed, with an adequate control arm, less than 20% of censored patients for PFS and no change in endpoint or sample size, accumulating three of five points for a high quality of clinical trial design, but did not meet its primary endpoint and failed in all criteria of the ESMO magnitude of clinical benefit. Therefore, in this metastatic, third line setting with a proportion of 95% of MMRp, neither atezolizumab monotherapy nor atezolizumab combined with cobimetinib can be recommended over SOC and further studies are warranted to elucidate their potential role in other settings.

### 2.6. Atezolizumab Combined with Chemotherapy as a First Line Treatment in Unresectable mCRC

Atezolizumab has also been tested as a combination partner with fluoropyrimidine-bevacizumab maintenance treatment (after induction with FOLFOX and bevacizumab) in unresectable, untreated BRAF^wt^ mCRCs [23]. However, when compared with the same treatment without atezolizumab after a median follow-up of 18.7 months, there was no improvement in PFS (HR = 0.96 (95% CI 0.77–1.20; *p* = 0.727)) nor OS (HR = 0.86 (95% CI 0.66–1.13; *p* = 0.28)).

## 3. Predictive Biomarkers

To date, the only predictor of response to checkpoint inhibitors is the presence of MMRd in the tumor. In the Checkmate study with nivolumab, none of the assessed biomarkers (PD-L1 expression, mutation status of the oncogenes *BRAF* and *KRAS*, history of Lynch syndrome) were predictive for response. In the case of pembrolizumab, the study from Le and colleagues [12] showed that a high number of mutations was associated with a longer PFS, while the presence of CD8+ lymphocytes and expression of PD-L1 showed a trend toward objective response, although differences in PFS and OS were not statistically significant. In the same study, a substantial decrease of the levels of the tumor marker ‘carcinoembryonic antigen’ (CEA) was only observed in MMRd CRCs and was predictive for both PFS and OS. There are preliminary results that CD8+/PD-1+ T-cell infiltration may be predictive of response in MMRp early CRC in the neoadjuvant setting, but this needs to be confirmed in larger trials. 

## 4. Quality of Life Indicators

The Checkmate 142 study assessed quality of life indicators using the EORTC-QLQ-C30 and the EQ-5D questionnaires. Results from the nivolumab in monotherapy or combination with ipilimumab were similar with more than 50% of patients maintaining functioning and global health without worsening of symptoms. Moreover, both schemes showed statistically significant and clinically meaningful improvements in functioning, symptoms and quality of life observed as early as week thirteen and were maintained for some indicators beyond week 37. Regarding the EQ-5D test, a clinically meaningful improvement of all five dimensions (mobility, self-care, usual activities, pain/discomfort and anxiety/depression) was observed also as early as week thirteen and was maintained during the treatment (up to 67 weeks in the combination arm).

## 5. Safety and Security

Overall tolerability of all checkpoint inhibitors was favorable with most of the adverse effects being mild (grade one and two) and not leading to treatment discontinuation nor withdrawal from the study. Since ICIs block a pathway that is thought to prevent autoimmunity, any relevant autoimmune precondition or immunosuppressive treatment were exclusion criteria as well as active or chronic hepatitis B/C or human immunodeficiency virus (HIV) infection. Drug related adverse effects (DRAE) grade three or four ranged from 20% to 41% and were mainly manageable. In the two Checkmate study arms, 6.8% and 13% discontinued treatment for any grade of adverse effect, respectively. The most frequent grade three and four adverse effects reported were fatigue, nausea/vomiting, anemia, lymphopenia, hyponatremia, colitis/diarrhea, gastritis/ulcer, arthritis/arthralgia, elevated liver enzymes, acute kidney injury and asymptomatic pancreatitis, most of which were either auto-limited, reversed after discontinuation or were treatable. Drug related adverse effects with potential immunologic etiology affected the skin (rash), liver, thyroid (hypothyroidism), GI tract (colitis), adrenal glands (adrenal insufficiency) or the lungs (pneumonitis). All were manageable with treatment discontinuation, short-term corticosteroid therapy or replacement therapy (in case of hypothyroidism). No drug related deaths occurred. In the Checkmate study, one sudden death occurred ten days after therapy discontinuation and under steroid therapy for colitis. After the autopsy, this death was not attributed to drug toxicity. All other reported deaths were caused by disease progression or other causes. Despite the overall excellent tolerability, anecdotally one case of pneumonitis was reported related to pembrolizumab and we found one case report in the literature of a severe necrotizing myositis associated with long term efficacy following nivolumab and ipilimumab combination therapy [24]. Regarding atezolizumab, patients had more grade 3–4 adverse effects when combined with cobimetinib compared to atezolizumab monotherapy, but a similar rate when compared with standard of care regorafenib (61, 31 and 58%, respectively) [22]. The most frequent adverse effects in the combination group were diarrhea (11%), anemia (6%), elevated creatine phosphokinase levels (7%) and fatigue (4%). Fatal events were rare and occurred in 3% of both the combination group (2 × sepsis) and the regorafenib group (1 × perforation), while there was no fatal event in the atezolizumab monotherapy group.

## 6. Future Studies

The results of the first studies with ICIs presented in this article have raised new hope for the treatment of CRC. As a consequence, numerous further studies are ongoing or planned to corroborate these data, test ICIs in different settings and investigate new ICI antibodies. Many ongoing studies seek to enhance treatment efficacy by combining ICIs with other therapeutic modalities such as chemotherapy (e.g., with pembrolizumab, NCT02375672, or with atezolizumab, NCT02912559), radiotherapy (e.g., with pembrolizumab, NCT02437071), or other targeted therapies such as regorafenib (with pembrolizumab, NCT03657641), binimetinib and bevacizumab (with pembrolizumab, NCT03475004), cetuximab (with pembrolizumab, NCT02713373), bevacizumab (with atezolizumab, NCT02982694) or cobimetinib (with ipilimumab and nivolumab, NCT02060188). Another focus is to assess ICI efficacy in the setting of microsatellite stable (MSS), mismatch repair proficient (MMRp) tumors, which were thought be naturally resistant to ICIs until Chalabi and colleagues showed this year that there might be a subset of patients that could benefit from them [19]. In this sense, studies are ongoing or planned that assess the usefulness of ICIs in MSS/MMRp patients particularly when combined with other treatments, for instance nivolumab with regorafenib (NCT04126733), nivolumab and ipilimumab combined with radiotherapy (NCT04575922) or nivolumab and regorafenib combined with radiotherapy (NCT04030260). For the combination of nivolumab and ipilimumab, Chalabi and colleagues also showed that ICIs could be very useful in the neoadjuvant setting. Trials are now on their way to determine the role of neoadjuvant pembrolizumab in mCRC (e.g., NCT03984578, NCT04231526). The neoadjuvant use of ICIs could also be interesting for rectal cancers following the rationale that radiotherapy increases the mutational burden and subsequent generation of neoantigens boosting the cytotoxic T-cell immune response against the tumor. All four ICIs are under investigation for that purpose: pembrolizumab (NCT04109755), nivolumab and ipilimumab (NCT04124601) and atezolizumab (NCT04017455, combined with bevacizumab). Another potentially useful approach is to pretreat MSS/MMRp tumors with temozolomide—an alkylating agent—to trigger a hypermutation status and make MSS/MMRp tumors more amenable to ICI treatment, for instance with pembrolizumab (NCT03519412) or nivolumab and ipilimumab (NCT03832621). Lastly, ICIs are also being investigated as combination partners for chemotherapy in the first line of treatment, e.g., nivolumab (NCT04072198) and atezolizumab (NCT03721653). In addition to the ICIs discussed so far in this article, new antibodies are on the horizon that have shown promising results in other cancers: durvalumab and avelumab (both anti-PD-L1) and tremelimumab (anti-CTLA-4). For instance, the combination of tremelimumab and durvalumab is actually under investigation in three different settings: in MSS mCRC after palliative radiotherapy (NCT03007407), as a combination partner for chemotherapy in the first line treatment of *KRAS* mutant mCRC (NCT03202758), and in advanced unresectable and treatment-refractory CRC in a randomized, open-label trial comparing with best supportive care (NCT02870920). Most of the ongoing or planned studies discussed here are small-to-medium sized, single arm, open-label phase two trials that seek to confirm objective responses, safety and tolerability. Thus, interpretation of efficacy, especially when compared to standard treatment, should be done cautiously. However, some phase three trials with adequate comparators are now on their way and the first results are expected within the next two years: standard of care chemotherapy (SOC) and nivolumab vs. SOC alone (NCT03414983), nivolumab and ipilimumab vs. nivolumab alone vs. SOC alone (NCT04008030) and SOC and atezolizumab vs. SOC alone (NCT02912559).

## 7. Search Strategy

We have used the following search terms in MEDLINE to find eligible studies: “Colorectal Neoplasms” (Mesh) AND (“immune checkpoint inhibitor” OR pembrolizumab OR nivolumab OR ipilimumab OR atezolizumab). Our selection strategy is displayed in Figure 1. Briefly, our research retrieved 194 results, 175 of which were excluded after screening for not being clinical trials. After full-text assessment, we further excluded 12 publications, six of which were phase I trials, which were not the focus of this article. In addition, we have screened conference proceedings of the American Society of Clinical Oncology (ASCO) and the ESMO and identified two additional unpublished studies. For the section of future studies, we have further screened the database clinicaltrials.gov using “colorectal cancer” as a search term for a condition or disease combined with one of the following: pembrolizumab, nivolumab, ipilimumab, atezolizumab, durvalumab, avelumab or tremelimumab. Results were filtered by recruitment status and studies marked as “suspended”, “terminated”, “completed”, “withdrawn” or with unknown status were excluded. For this section, results were handpicked according to our consideration of relevance.

## 8. General Recommendation

Immunotherapy based on ICIs in monotherapy (anti-PD-1) or combination therapy (anti-PD-1 and anti-CTLA-4) should be offered as a second line therapy to mCRC patients with tumors that display MMRd demonstrated by either microsatellite instability and/or by immunohistochemistry (loss of expression of MLH1, MSH2, MSH6 or PMS2). Unpublished data suggest moderate evidence that pembrolizumab may be superior to standard of care chemotherapy as a first line treatment in MMRd mCRC and therefore should be considered as an option. There is preliminary evidence that ICIs may also be helpful in the neoadjuvant setting of early tumors, but this needs to be confirmed in larger future trials.

## Figures and Tables

**Figure 1 jcm-09-03520-f001:**
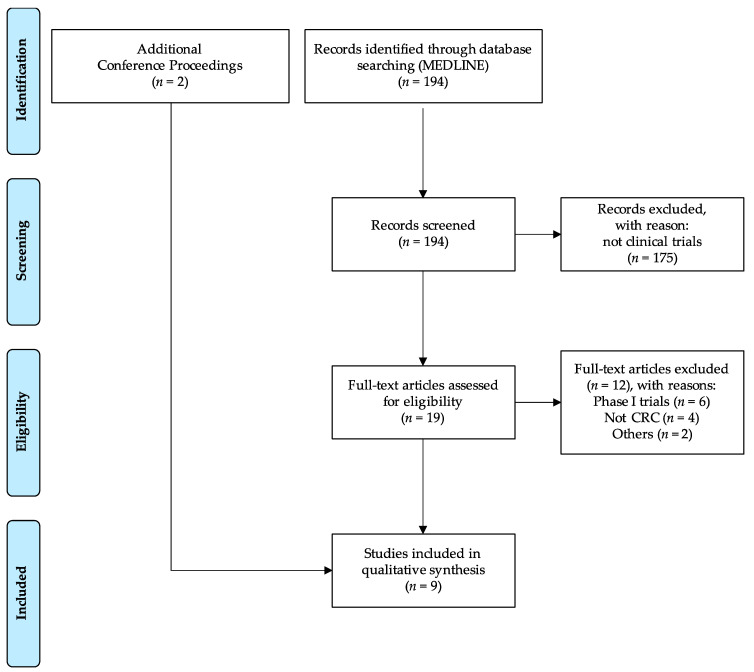
Consort flow diagram of studies reviewed; CRC, colorectal cancer.

**Table 1 jcm-09-03520-t001:** Published and/or presented trials with checkpoint inhibitors in metastatic colorectal cancer.

Reference	Checkpoint Inhibitor	Design & Cohorts (Treating Arms)	Dosing	Median Follow-Up	Primary (1) and Secondary Endpoints (2)	ORR	PFS	OS	Most Important Adverse Effects (DRAE)	Quality of Life Indicators	Quality of Trial Design	ESMO-MCBS
**Le DT, NEJM, 2015** **NCT01876511**	Pembrolizumab (Phase II)	All treatment refractory progressive metastatic cancers (ECOG PS 0–1). Cohort 1: 11 MMRd mCRC Cohort 2: 21 MMRd mCRC Cohort 3: 9 MMRd not CRC CRCs with ≥2 previous chemotherapy regimens All ECOG 0–1.	Pembro 10 mg/kg Q2W	Cohort 1: 36 w Cohort 2: 20 w Cohort 3: 21 w	Immune related ORR (1) Immune related PFS (1)	40% in MMRd mCRC vs. 0% in MMRp mCRC (95% CI: 12–74) CR: 0% PR: 40% SD: 50% DCR: 90% 71% in MMRd non-mCRC	78% in MMRd mCRC vs. 11% in MMRp mCRC (at 20 weeks). In MMRd mCRC the median PFS was not reached; in MMRp 2.2 mts. (HR 0.10, *p* < 0.001)	In MMRd mCRC median OS was not reached; in MMRp 5.0 mts. (HR = 0.22, *p* = 0.05)	41% had grade 3 or 4 AE. Most frequent: anemia, lymphopenia, asymptomatic pancreatitis, elevated liver enzymes, hyponatremia, hypoalbuminemia, bowel obstruction. One case of pneumonitis (2%).	Not reported	3/5 (high)	2/3 (strong)
**Le DT, Science, 2017** **NCT01876511**	Pembrolizumab (Phase II)	86 patients with 12 different tumor types (40 CRCs), all treatment refractory progressive and metastatic MMRd (ECOG PS 0–1). 81% had received ≥2 previous treatments. 48% confirmed to have Lynch syndrome. Radiographic assessment at 12 w, then every 8 w the first year and then every 12 w.	Pembro 10 mg/kg iv Q2W	12.5 months	Immune related ORR (1) Immune related PFS (1)	52% for mCRC (95% CI: 36–68) and 54% for other cancers (39–69). For mCRC: · CR: 12% · PR: 40% · SD: 30% · DCR: 82%. No difference between Lynch and non-Lynch.	Median PFS was not reached for mCRC (estimates at 1 yr. = 64%; at 2 yrs. = 53%).	Median OS was not reached (estimates at 1 yr. = 76%; at 2 yrs. = 64%).	20% grade 3 or 4 AE. Most frequent: asymptomatic pancreatitis, diarrhea/colitis, anemia, fatigue, arthritis/arthralgia. 21% hypo-thyroidism treated with supplement therapy.	Not reported	2/5 (low)	2/3 (strong)
**Díaz L, ESMO 2017 Conference, Ann Oncol, NCT02460198** **“Keynote 164” trial** **“Keynote 158” trial**	Pembrolizumab (Phase II)	KN164 cohort: 61 patients with MSI-H mCRC and ≥2 prior therapies KN158 cohort: 77 patients with MSI-H non-CRC and ≥1 prior therapies Response was assessed every 9 w.	Pembro 200 mg Q3W		ORR (1)	ORR for mCRC: 27.9% (95 CI%: 17.1%–40.8%) · ORR for non-CRC: 37.7% (95% CI 26.9%–49.4%)	43% (at 6 mts) for mCRC 45% (at 6 mts) for non-CRC	87% (at 6 mts) for mCRC 73% (at 6 mts) for non-CRC Median OS was not reached.	7% (mCRC) and 9% (non-CRC) had serious drug related AE.	Not reported	2/5 (low)	2/3 (strong)
**Overman MJ, Lancet Oncology, 2017** **NCT02060188** **Part of the “Checkmate 142” study**	Nivolumab (Phase II)	Multicenter, open label, no control group, non-randomized, 74 patients with MMRd metastatic or recurrent CRC (ECOG PS 0–1). 85% had received ≥2 lines of previous ttm. Follow-up for 3 yrs. Treatment until disease progression, death, unacceptable toxic effects or withdrawal from study.	Nivol 3 mg/kg Q2W	12 months	IA-ORR (1) BICR-ORR (2)	IA-ORR: 31.1% (20.8–42.9) CR: 0% PR: 31.1% SD: 37.8% BICR-ORR: 32.4% (95% CI: 22.0–44.3) CR: 2.7% PR: 29.7% SD: 33.8%	50.4% (at 1 yr.) 95% CI: 38–61 Median PFS was not reached.	73.4% (at 1 yr) 95% CI: 62–82 Median OS was not reached.	Drug-related AE: 48.6% grade 1 or 2 AE. 17.6% grade 3 AE. 2.7% grade 4 AE. 1.4% grade 5 AE. 8% lipase elevation and 3% amylase elevation = the only grade 3/4 AE. Only 6% discontinued due to AE.	At week 13, meaningful improvements in functioning, symptoms and QoL, with some maintained through week 37 or beyond. (Assessing tools: EORTC QLQ-C30 and EQ-5D)	2/5 (low)	2/3 (strong)
**Overmann MJ, J Clin Oncol, 2018** **NCT02060188** **Part of the “Checkmate 142 study”**	Nivolumab + Ipilimumab (Phase II)	Multicenter, open label, no control group, non-randomized, **119** patients with MMRd metastatic or recurrent CRC (ECOG PS 0–1). 76% had received two or three lines of previous treatment. Treatment until disease progression, discontinuation, death, withdrawal of consent or study end.	Nivol 3 mg/kg plus Ipi 1 mg/kg Q3W (four doses) followed by nivol 3 mg/kg Q2W	13.4 months	IA-ORR (1) BICR-ORR (2)	IA-ORR: 54.6% (45.2–63.8) CR: 3.4% PR: 51.3% SD: 31% BICR-ORR 49%(95% CI: 39.5–58.1) CR: 4.0% PR: 45% SD: 33%	· 71% (at 1 yr.) (95% CI: 61.4–78.7) · Median PFS was not reached.	85% (at 1 yr.) (95% CI: 77.0–90.2) Median OS was not reached.	Drug-related AE: · 32% grade 3 or 4 (all manageable): elevated transaminases, lipase, anemia and colitis. Discontinuation for any grad: 13%. AEs with potential immunologic etiology: 29% (skin), 25% (endocrine), 23% (GI), 19% (hepatic), and 5% (pulmonary, renal).	At week 13, meaningful improvements in functioning, symptoms and QoL. (Assessing tools: EORTC QLQ-C30 and EQ-5D)	2/5 (low)	2/3 (strong)
**Eng C, Lancet Oncol, 2019** **NCT0278879**	Atezolizumab ± Cobimetinib (Phase III)	Cohort 1: 183 Atezolizumab + Cobimetinib Cohort 2: 90 Atezolizumab (alone) Cohort 3: 90 Regorafenib (SOC) Unresectable, locally advanced or mCRC with at least 1 prior treatment. ECOG PS 0–1	Ate 840 mg iv Q2W + Cob 60 mg po QD day 1–21 or Ate 1200 mg iv Q3W or Rego 160 mg po QD day 1–21	7.3 months	OS (1) PFS, ORR, DoR (2)	No CR in any of the cohorts	Median PFS: Cohort 1: 1.91 Cohort 2: 1.94 Cohort 3: 2.0 HR (1 vs. 3): 1.25 (95% CI 0.9–1.7) HR (2 vs. 3): 1.39 (95% CI 1.0–1.9)	Median OS: Cohort 1: 8.9 Cohort 2: 7.1 Cohort 3: 8.5 HR (1 vs. 3) 1.00 (95% CI 0.7–1.4) HR (2 vs. 3): 1.19 (95%CI 0.8–1.7)	100% had AEs. Grade 3–4 AE in cohort 1, 61%; cohort 2, 31%; cohort 3, 58%. Most frequent AE in cohort 1: diarrhea (11%), anemia (6%), elevated CK (7%), fatigue (4%). Serious AE in cohort 1, 40%; cohort 2, 17%; cohort 3, 23%.	Not reported	3/5 (high)	0/3 (low)
**Le DT, J Clin Oncol, 2020** **NCT02460198** **“Keynote 164” update**	Pembrolizumab (Phase II)	Cohort 1: 61 MMRd CRC ≥2 lines of treatment Cohort 2: 63 MMRd CRC ≥1 lines of treatment Unresectable, locally advanced or metastatic CRC. ECOG PS 0–1	Pembro 200 mg iv Q3W	Cohort 1: 31.3 mts Cohort 2: 24.2 mts	ORR (1) PFS, OS, DoR, safety, tolerability (2)	Cohort 1: 33% (95% CI: 21–46) CR: 3.3% PR: 29.5% SD: 18.0% DCR: 50.8% Cohort 2: 33% (95%CI: 22–46) CR: 7.9% PR: 25.4% SD: 24% DCR: 57.1%	Cohort 1: median 2.3 mts (95% CI 2.1–8.1) 34% at 1 yr. 31% at 2 yrs. Cohort 2: median 4.1 mts (95% CI 2.1–18.9) 41% at 1 yr. 37% at 2 yrs.	Cohort 1: median 31.4 mts (95% CI 21.4 mts-NR) 72% at 1 yr. 55% at 2 yrs. Cohort 2: median NR (95% CI 19.2-NR) 76% at 1 yr. 63% at 2 yrs.	Drug-related AE: Cohort 1: grade 3–4: 3% fatigue, 2% asthenia Cohort 2: grade 3–4: None Immune-mediated AE Cohort 1: 21% (7% grade 3–4); pancreatitis, hepatitis, pneumonitis, skin toxicity. Cohort 2: 37% (3% grade 3–4); colitis, pneumonitis. 2% in each cohort led to discontinuation (pneumonitis in both cohorts) No grade 5 AE in both cohorts.	Not reported	2/5 (low)	2/3 (strong)
**Chalabi M, Nature Medicine, 2020** **NCT03026140**	(Nivolumab + Ipilimumab) ± Celecoxib (Phase II)	Cohort 1: 20 MMRd CRC Cohort 2: 15 MMRp CRC Resectable early stage (I, II or III). All ECOG 0–1.	Nivol 3 mg/kg D1 + D15 + Ipilimumab 1 mg/kg D1 ± Celecoxib 200 mg QD from D1 until surgery	9.0 months	Safety & feasibility (1)	Cohort 1: 100% (95% CI 86–100) CR: 60% (12/20) MPR: 95% (19/20) Cohort 2: 27% (95% CI 8–55) CR: 13.3% (2/15) MPR: 20% (3/15) PR: 6.7% (1/15)	Not reached	Not reached	All patients could undergo surgery. 13% grade 3–4 drug related AE. 2 patients grade 3 rash (resolved with corticosteroid therapy). One patient had grade 3 colitis (resolved 3 d after infliximab SD). Three patients grade 3 asymptomatic laboratory test. Eight patients grade 3 surgery related AE-> 4/8 were anastomotic leakages ->1/4 had complete response and showed signs of colitis (probably drug-related).	Not reported	2/5 (low)	NA
**Andre T, JCO, 2020 NCT02563002**	Pembrolizumab (Phase III)	Open label, randomized. First line treatment of MMRd mCRC Cohort 1: 153 Pembrolizumab Cohort 2: 154 Standard Chemotherapy (mFOLFOX6 or FOLFIRI Q2 W ± bevacizumab or Cetuximab). ECOG PS 0–1.	200 mg Pembro Q3W for up to 2 years	Cohort 1: 28.4 mts Cohort 2: 27.2 mts	PFS, OS (1) ORR, safety (2)	Cohort 1: 43.8% Cohort 2: 33.1%	Median PFS: Cohort 1: 16.5 mts Cohort 2: 8.2 mts HR: 0.60; (95% CI 0.45–0.80), *p* = 0.0002 12 mts PFS rate: 55.3 vs. 37.3% 24 mts PFS rate: 48.3 vs. 18.6%	Not reached	Drug related AE grade 3–5 Cohort 1: 22% Cohort 2: 66% (One pt. in cohort 2 (chemo) died because of DRAE.)	Not reported	3/5 (high)	NA

Durable response: ≥ 12 weeks. Centralized results are reported; MMRd, mismatch repair deficient; MSI-h, microsatellite instability high; HR, hazard ratio; CI, confidence interval; CRC, colorectal cancer; mCRC, metastatic CRC; rCRC, recurrent CRC; D, day; w, weeks; mts, months; QD, every day; Q2W, every two weeks; Q3W, every three weeks; ORR, objective response rate; IA-ORR, investigator assessed ORR; BICR-ORR, blinded independent central review assessed ORR; PFS, progression free survival; OS, overall survival; CR, complete response; PR, partial response; MPR, major pathological response (≤10% viable tumor); SD, stable disease; DC, disease control (= CR + PR + SD); MPR, major pathologic response; DCR, disease control rate; DoR, duration of response; AE, adverse effects; QoL, quality of life; NR, not reported; NA, not applicable; ESMO-MCBS, see text; ECOG PS, Eastern Cooperative Oncology Group Performance Status.

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
