# Peer review of "Current Treatments of Metastatic Colorectal Cancer with Immune Checkpoint Inhibitors—2020 Update"

_jcm, 2020, doi:10.3390/jcm9113520_

Round 1

Reviewer 1 Report

Overall this is a clear, cosncise and well written manuscript.  Sufficient critical evaluation of  the data available from the previous published clinical trials  are presented for readers. It was made a systematic contribution to the
research literature in this area of investigation. Only a minor comment follow:

Introduction, p2, line 50: Suggest to provide the assay for MSI testing, for instance "fragment analysis by capillary electrophoresis or by next generation sequencing".

Author Response

Response to Reviewer 1 Comments

Point 1: Overall this is a clear, concise and well written manuscript. Sufficient critical evaluation of the data available from the previous published clinical trials are presented for readers. It was made a systematic contribution to the research literature in this area of investigation. Only a minor comment follow:
Introduction, p2, line 50: Suggest to provide the assay for MSI testing, for instance "fragment analysis by capillary electrophoresis or by next generation sequencing".

Response 1: We are very glad that you enjoyed reading our manuscript and want to thank you for your time and comments. We agree with you that MSI testing is indeed an important issue, since there are still inconsistencies between clinics and practitioners on when and how to perform it. Therefore, we’ve decided to include an additional sentence that points to current guidelines (most recently ESMO 2019 and BSG 2020).
We also mention the two accepted methods for MSI testing – PCR and NGS – without going into detail (e.g. discussing specific panels) because we don’t consider it the scope of our article. Please find the additional sentence in the revised manuscript on page 2, line 103.

Reviewer 2 Report

This is a very interesting review articles providing information of immune checkpoint inhibitors in colorectal cancer.

There are some issues which need to be addressed.

  1. The authors should mention primary and secondary endpoint of phase 2 and 3 studies in the main text and tables
  2. Line 14: delete adjuvant
  3. Line 15: “However, a vast majority of metastatic cases are resistant to conventional chemotherapy” this sentence is not correct. Objective response rates of 1st line FOLFOX, FOLFIRI or FOLFOXIRI is 50-60%.
  4. Line 42: “if an inhibitory signal…”. Please re-write this sentence. CTLA4 and CD28 competes for ligands CD80/86 on APCs and the binding of CTLA4 to CD80/86 induces inhibitory signals
  5. Line 98: In Keynote 164 study, cohort A was ≥2 previous lines of systemic therapy and B was ≥1 lines. However, 10% pts had 1 line in cohort A and 38% pts had 1 line in cohort B. It should be mentioned.
  6. Line 99: delete “adjuvant” it was not adjuvant chemo
  7. Line 116: “the final result” > should be “interim analysis”
  8. Line 126: “before randomization” > “after”
  9. Line 221: “3.1 subsection” ??
  10. Line 260: Please add adjuvant trials including NCT02912559

Author Response

Response to Reviewer 2 Comments

This is a very interesting review articles providing information of immune checkpoint inhibitors in colorectal cancer.

There are some issues which need to be addressed.

Thank you very much for your time. We appreciate that you found our article interesting and want to thank you for your comments that help us to further improve our manuscript. Please find our detailed answers below.

Point 1: The authors should mention primary and secondary endpoint of phase 2 and 3 studies in the main text and tables.

Response 1: We agree with you that defining the endpoints is not only a sign of highquality science, but also necessary when it comes to globally assess the outcomes of a study. Most of the studies were phase II trials with comparable designs and very similar primary and secondary endpoints (ORR or PFS). For that reason, we omitted this detail also to enhance readability. However, since it might be relevant to some readers, we have now added these details in the main text and have dedicated to it an additional column in the table.

Point 2: Line 14: delete adjuvant.

Response 2: Thank you for this hint. Indeed, since in this sentence we talk about colorectal cancer, it was not only adjuvant chemotherapy that has improved survival of colon cancer, but also neoadjuvant chemotherapy that has improved survival of rectal cancer. Hence, we’ve deleted ‘adjuvant’.

Point 3: Line 15: “However, a vast majority of metastatic cases are resistant to conventional chemotherapy” this sentence is not correct. Objective response rates of 1st line FOLFOX, FOLFIRI or FOLFOXIRI is 50-60%.

Response 3: We agree that this sentence is equivocal. What was meant is that even though tumors respond to first line treatment as you are saying correctly (50-60% partial response), in our experience many of them will progress after a PFS of about 9-

12 months, so become resistant requiring further lines of treatment. To avoid this ambiguity, we have reformulated the sentence (p. 1, line 58).

Point 4: “if an inhibitory signal…”. Please re-write this sentence. CTLA4 and CD28 compete for ligands CD80/86 on APCs and the binding of CTLA4 to CD80/86 induces inhibitory signals.

Response 4: We have reformulated and clarified the mechanisms of how CTLA-4 acts on

T cells. Please find the reformulated sentence on page 1, line 85.

Point 5: Line 98: In Keynote 164 study, cohort A was ≥2 previous lines of systemic therapy and B was ≥1 lines. However, 10% pts had 1 line in cohort A and 38% pts had 1 line in cohort B. It should be mentioned.

Response 5: Indeed, a good point. Different to the authors of the original article we have renamed the two cohorts in ‘less’ and ‘more’ previous lines of treatment, since it is confusing to talk about ‘≥2 lines’, when there is a not negligible 10% of only 1 previous line. Please find the corrections on page 3, line 162).

Point 6: Line 99: delete “adjuvant” it was not adjuvant chemo

Response 6: Thank you for the hint. Since the cohorts were comprised of metastatic cancers who had not undergone surgery, of course it was not ‘adjuvant’ treatment. Please find the correction on page 3, line 162.

Point 7: Line 116: “the final result” > should be “interim analysis”

Response 7: Thank you for the hint. Indeed, what was meant was the interim analysis of the final PFS analysis – and not the final results. Please fin the correction now on page 3, line 180.

Point 8: Line 126: “before randomization” > “after”

Response 8: We have revised the abstract and found that the chemotherapy was chosen prior to randomization. You can verify this here:

https://ascopubs.org/doi/10.1200/JCO.2020.38.18_suppl.LBA4

Point 9: Line 221: “3.1 subsection” ??

Response 9: Has been deleted.

Point 10: Line 260: Please add adjuvant trials including NCT02912559

Response 10: Initially, our intention and main focus was to inform and guide the reader through the actual available evidence and not to confuse with too many unpublished trials. However, it is true that treatment with ICI is a rapidly evolving field and we agree with you that some ongoing or planned trials are worth to be mentioned. For that reason, we have added a whole new paragraph that does not claim to be complete but gives an overview about future trials that we consider could answer relevant open questions with ICI treatment, including ICI as combination partners in adjuvant and neoadjuvant settings. Please find this new paragraph on page 6 from line 331 on.

Reviewer 3 Report

This is an interesting review that summarized the current treatments of metastatic colorectal cancer  with immune checkpoint inhibitors. Overall, the manuscript is well written; please revise the English language and check throughout the text for spelling errors (i.e. line 221: please delete: “3.1. Subsection”). Perhaps, further discussion could be added in the paragraph about future studies.

Author Response

Response to Reviewer 3 Comments

This is an interesting review that summarized the current treatments of metastatic colorectal cancer with immune checkpoint inhibitors. Overall, the manuscript is well written; please revise the English language and check throughout the text for spelling errors (i.e. line 221: please delete: “3.1. Subsection”).

We thank you for your time and are glad that you enjoyed reading our manuscript. We hope that we have now corrected all spelling errors.

Point 1: Perhaps, further discussion could be added in the paragraph about future studies.

Response 1: Initially, our intention and main focus was to inform and guide the reader through the actual available evidence and not to confuse with too many trials. However, it is true that treatment with ICI is a rapidly evolving field and we agree with the reviewer that some ongoing or planned trials are worth to be mentioned. For that reason, we have added a whole new paragraph that does not claim to be complete but gives an overview about future trials that we consider could answer relevant open questions with ICI treatment. Please find this new paragraph on page 6 from line 331 on.

Reviewer 4 Report

This is a valuable paper because many clinicians are still unfamiliar to true efficacy of ICI to mCRC.

The author should revise or add as follows;

  1. The author should describe the search strategy on the research and the references clearly.
  2. Consort diagram that selects the research is necessary.

Author Response

Response to Reviewer 4 Comments

This is a valuable paper because many clinicians are still unfamiliar to true efficacy of ICI to mCRC.

Thank you very much for your time. We appreciate that you find our paper valuable.

The author should revise or add as follows:
Point 1: The author should describe the search strategy on the research and the references clearly.
Point 2: Consort diagram that selects the research is necessary.

Response 1 & 2: Our article is written in the style of a narrative review article and the real evidence on the topic is still not very abundant. For that reasons, initially we did not include a paragraph on the search strategy nor a consort diagram. However, we agree with you that it is always a sign of good science and for some readers it might add valuable information. Hence, we´ve decided to follow your recommendation. Please find the corresponding paragraph under section 7, on page 7, starting from line 373, as well as the consort diagram at the end of the article.

Round 2

Reviewer 2 Report

  1. line 41: "apoptosis in regulatory T cells." the role of PD-1 on Tregs is very controversial (Nature immunology 21, 1346-1358 (2020)). Please delete regulatory T cells or describe  all  possible role of PD-1 on Tregs
  2. line 219 : "3. predictive biomarkers" Based on Keynote 158, FDA approved pembrolizumab for metastatic solid cancer (including colorectal cancer) with high tumor mutation burden. Although there was no colorectal cancer in Keynote 158, there are several reports showing high tumor mutation burden without MMR deficiency in colorectal cancer (Oncologist. 2019;24(10):1340. Epub 2019 Apr 30). high tumor mutation burden should be added in "predictive biomarkers".
  3. line 229: please delete "3.1.subsection"
  4. line 240: "5.safety and security" please put checkmate number and add references